# Single-Cell Labeling Strategies to Dissect Neuronal Structures and Local Functions

**DOI:** 10.3390/biology12020321

**Published:** 2023-02-16

**Authors:** Keigo Kohara, Masayoshi Okada

**Affiliations:** 1KMU Biobank Center, Institute of Biomedical Science, Kansai Medical University, Hirakata 573-1010, Japan; 2Department of Medical Life Science, College of Life Science, Kurashiki University of Science and the Arts, Kurashiki 712-8505, Japan

**Keywords:** single-cell labeling methods, neuron, synapse, morphology, neural circuit, cre recombinase, single-cell gene knockout, fluorescent protein, neuronal activity, single-cell silencing and activation, single-cell activity manipulating analysis

## Abstract

**Simple Summary:**

The neuronal circuits are essential for memory, recognition, and behavior but are too complex to be examined with ordinary, global labeling methods. Moreover, the role of each protein/gene in forming circuits is one of the biggest open questions in modern biology. Single-cell labeling methods are potent approaches for examining neuronal structure and circuits. Single-cell transgenic methods also enable single-cell gene knockout and modulation of neuronal activity, which can reveal their functional roles. This review summarizes the details of single neuronal labeling methods of non-transgenic and transgenic strategies and their contributions to our understanding of neuronal structures and functions.

**Abstract:**

The brain network consists of ten billion neurons and is the most complex structure in the universe. Understanding the structure of complex brain networks and neuronal functions is one of the main goals of modern neuroscience. Since the seminal invention of Golgi staining, single-cell labeling methods have been among the most potent approaches for dissecting neuronal structures and neural circuits. Furthermore, the development of sparse single-cell transgenic methods has enabled single-cell gene knockout studies to examine the local functions of various genes in neural circuits and synapses. Here, we review non-transgenic single-cell labeling methods and recent advances in transgenic strategies for sparse single neuronal labeling. These methods and strategies will fundamentally contribute to the understanding of brain structure and function.

## 1. Introduction

The rodent brain consists of one hundred million neurons that form complex neural networks via synaptic connections. The brain performs essential functions pertaining to memory, recognition, behavior, and survival. Understanding the structure and function of the rodent brain is one of the main goals of neuroscience research. 

In 1873, Camillo Golgi developed the non-transgenic single cell-labeling Golgi staining method [1], which enabled the sparse and random visualization of the morphology of single neurons. For more than 100 years, Golgi staining has been a standard method for dissecting the morphological structures of neurons and neuronal networks [1,2,3]. Although Golgi staining is a non-transgenic method, it is one of the seminal methods in the history of life science. However, it does have several experimental limitations, such as a lack of cell-specificity and multicolor staining, and the impossibility of live imaging or examining relationships with functionalities [3]. From the 1950s to the 1970s, revolutionary progress was achieved in molecular biology and molecular genetics [4]. Then, the method for generating transgenic mice [5] and the first gene knockout mouse were developed [6]. Furthermore, the discovery of green fluorescent proteins (GFPs), multiple fluorescent proteins, and their transgenic applications in rodents have induced revolutionary progress in biology with fluorescent proteins [7,8]. Since these landmark studies, various transgenic methods have been developed in the field of life sciences and neuroscience using rodent model animals [9,10]. Of these, single-cell labeling methods are among the most powerful tools for dissecting complex neuronal synaptic networks and their local functions. Furthermore, single-cell gene knockout strategies were developed to dissect local functions of neural circuits [11,12]. In addition, sparse single-cell transgenic methods enable the manipulation of local neuronal activity to understand activity-dependent mechanisms of neuronal circuit formation [13]. In this manuscript, we review the current progress in the field of neuroscience, with a focus on the strategies and methods in single-cell labeling, single-cell gene knockout, and single-cell activity manipulation analyses.

## 2. Non-Transgenic Methods for Sparse Single-Cell Labeling

For Golgi staining, brain slices are fixed with chromic and osmic acids. The brain slices are then incubated with solutions containing silver nitrate (Figure 1A) [3]. After these treatments, single neurons are sparsely visualized in the brain slices (Figure 1A,C) [3]. An example of the successful application of Golgi staining is the discovery of synapses by Ramon y Cajal [2]. This has been one of the most significant discoveries in neuroscience [2]. Nevertheless, Golgi staining has several experimental limitations, such as the lack of cell-specificity and multicolor staining, incompatibility for combinational use with immunohistochemistry, and the impossibility of live imaging or examining relationships with functionalities [3]. 

Various single-cell labeling methods have been developed to overcome these limitations (Figure 1C and Figure 2). These single-cell labeling methods comprise both non-transgenic (Figure 1C) and transgenic methods (Figure 2). Non-transgenic methods of single-cell labeling are mainly based on the microinjection of traceable tag molecules inside cells using a sharp glass micropipette electrode, application of physical pressure, or diffusion. Horseradish peroxidase (HRP), biocytin, neurobiotin, biotinylated dextran amine, and various fluorescent dyes (including lucifer yellow) are microinjected for visualizing single-cell morphologies (Figure 1B,C) [12,14,15,16,17,18,19,20,21,22,23,24,25]. These non-transgenic single-cell labeling methods have been applied to dissociated cultured neurons, acute brain slices, cultured brain slices, and neurons in vivo [12,14,15,16,17,18,19,20,21,22,23,24,25]. The signals of traceable tag molecules are specifically amplified using enzymatic reactions or secondary antibodies with traceable tags or biotin–avidin complexes (Figure 1B,C). Unlike Golgi staining, microinjections of traceable tags can be used for immunohistochemical staining. Thus, they enable the examination of endogenous protein distributions in neuronal structures, such as spines or axonal terminals. The relationship between cell morphology and cellular diversities can also be examined using antibodies that are specific to synaptic marker proteins and marker proteins of cellular diversities [12,14,15,16,17,18,19,20,21,22,23,24,25].

Furthermore, microinjections of traceable tags can be performed in conjunction with in vitro and in vivo electrophysiology, such as patch clamp recording, intracellular recording, and juxta-cellular recording [12,14,15,22,23,24,25,26]. These advantages enable the examination of the relationship between cellular diversities and functionalities or morphologies and functionalities by combining microinjections of traceable tags with various electrophysiological methods. A seminal example of applying this technique is the discovery of cell- and subclass-specific firing patterns of hippocampal interneurons using electrophysiological microinjections of biocytin [22,23,24,25]. 

In addition, the Diolistc labeling method using a gene gun and fluorescent dyes also enables the sparse visualization of single neuron morphologies (Figure 1C) [27].

## 3. Physical Transgenic Methods for Sparse Single-Cell Labeling

As a result of advanced developments in the field of molecular biology and genetics [4], various transgenic strategies and methods have been developed for single-cell labeling (Figure 2). Physical transgenic methods using microinjection or biolistic gene gun have been developed as simple transgenic methods for single-cell labeling [12,28,29,30,31]. 

Microinjection methods use sharp glass micropipettes or electrodes to deliver DNA plasmids, which constitutively express fluorescent proteins such as GFP into the nuclei of single neurons in vitro (Figure 2) [29,30,31]. Previously, neurotrophic factors were considered “absolute retrograde acting molecules.” However, with the success of single-cell transgenic microinjection, this strong dogma has been drastically revised [29]. At the same time, brain-derived neurotrophic factor (BDNF) is transferred to cortical postsynaptic neurons in an activity-dependent manner [29].

In contrast, biolistic methods use gene guns to deliver DNA plasmids of fluorescent proteins using physical air pressure and gold particles (Figure 2) [12,28]. As both methods sparsely deliver transgenes into single cells among many non-transfected cells, the morphologies of single sparse neurons, such as dendrites, axonal terminals, and spines, can be clearly visualized [12,28,29,30,31]. Using a biolistic gene gun, brain BDNF was found to regulate the dendritic growth of cortical neurons [28]. Microinjection methods have been applied to dissociated neuronal cultures and organotypic slice cultures in vitro and neurons in vivo. Biolistic methods are mainly applied to dissociated neuronal and organotypic slice cultures in vitro.

## 4. Sparse Single-Cell Labeling Using Electroporation Methods and by Injection of Diluted Viruses

In utero electroporation is a transfection method that delivers transgene plasmids into embryos using electrodes [32,33,34,35,36,37,38]. Transgenic strategies that use in utero electroporation enable sparse single-cell labeling [39,40]. Other methods using electroporation can also induce sparse single-cell visualization [41,42,43,44]. An example of researchers applying sparse single-cell labeling using in utero electroporation was the discovery that two distinct cell groups with different birthdates in the cerebral cortex send differential axonal projections [45]. However, skilled technicians are required to perform the electroporation.

Injections of diluted transgenic viruses expressing fluorescent proteins also enable sparse single-cell labeling in the brain [46,47,48,49,50]. Using diluted transgenic sindbis viruses, projection patterns and dendritic morphologies of the posterior thalamic nuclei were uncovered [51].

However, as some viruses are known to have toxicities towards neurons, experimental time windows are limited in some cases [46,49]. 

## 5. Photoactivatable Fluorescent Proteins Mediated Sparse Single-Cell Labeling

Since the discovery of GFP, various types of fluorescent proteins have been developed [7,8]. Photoactivatable fluorescent proteins have interesting characteristics, such as ultraviolet (UV) light-activated enhancement of fluorescence [52,53]. Nevertheless, photoconvertible fluorescent proteins also have unique characteristics that show conversion of the fluorescence wavelength after illumination with light of specific wavelengths [54,55,56,57]. Both types of fluorescent proteins can be used for single-cell labeling using local illumination with light of specific wavelengths (Figure 2). For example, transgenic mice were generated with the ability to express Kaede, which is a photoconvertible fluorescent protein, enabling the monitoring of cell movement of various cells in the entire mouse body using photoconvertible labeling [58].

However, long-term illumination with UV light is necessary for visualizing the entire cell morphology. 

## 6. Photoactivatable and Drug-Activatable Cre Recombinase Mediated Sparse Single-Cell Labeling

Cre recombinase is a major genetic tool in the field of life science. 

Recently, photoactivatable Cre recombinase systems were developed. In these systems, Cre recombinases are split into two fragments: N-terminal and C-terminal [59,60,61,62,63,64]. Then, the N- and C-termini of Cre recombinase are fused with light-sensitive proteins that induce dimerization after light illumination (e.g., cryptochrome 2, CIB1, Magnets, and Vivid), which then induces recombination and the Cre-dependent expression of transgenes [59,60,61,62,63,64]. On the other hand, single-chain photoactivatable Cre recombinase was also developed by using the LOV domain of protein VVD [65]. Therefore, local and transient illumination of light-induced sparse recombination and sparse single-cell labeling with fluorescent proteins is achieved (Figure 2). Recently, transgenic mice constitutively expressing photoactivatable Cre recombinase have been generated [66]. In fact, applying sparse single-cell labeling to the entire mouse body is possible [66].

To achieve drug-induced sparse single-cell labeling, the Cre recombinase-fused estrogen receptor Cre-ER was used [67,68,69,70]. Cre-ER is activated using tamoxifen treatment. The transient applications of tamoxifen-induced sparse single-cell labeling in the brain region (Figure 2) include the use of Cre-ER-mediated sparse single-cell labeling to visualize the memory engram cells, which helped discover the functionally distinct memory engram cell groups in the brain [71]. 

## 7. A Leaky Expression-Dependent Sparse Single-Cell Labeling Method: Supernova

The TetO promoter is activated by tTA and is known to exhibit weak, leaky expression in the absence of tTA [72,73]. Supernova is a transgenic system that was developed using the leaky Cre recombinase expression from the tetO promoter [35,74]. In this system, the leaked expression of Cre recombinase is followed by an amplified feedback expression of tTA. This combination of leaky and feedback expression induces sparse single-cell labeling (Figure 2). Using Supernova, the dendritic morphologies of cortical layer four neurons were found to be regulated by NMDA receptor activation during developmental reorganization of thalamocortical connectivity [74]. The Supernova method has the advantage of a high level of transgene expression because of its feedback system. Furthermore, combining the Supernova system with retrograde viruses enabled projection-specific sparse-single-cell labeling [75]. 

## 8. A multicolor Sparse Single-Cell Labeling Method: BATTLE 2.0

Previously, developing methods that enable the split-tunable expression of transgenes was challenging. To overcome this problem, a strategy called BATTLE was developed by applying the concept of battle of transgenes [76,77]. In the BATTLE method, multiple recombinases are genetically designed to compete with each other and induce split-tunable expression of transgenes. Furthermore, BATTLE 2.0 was developed using battles of triple recombinase, Cre, FLPO, and Dre (Figure 2). It enabled multicolor and mutually exclusive sparse single-cell labeling and strong transgene expression, which visualizes the morphologies of dendrites and axonal terminals of single neurons (Figure 3) [76].

Although visualizing the structure of the entire synapses using light has been challenging, the BATTLE method enables high-resolution imaging of whole structures of both pre- and post-synapses in the mouse hippocampus with simultaneous visualization of endogenous synaptic proteins [76,77]. 

## 9. MADM, MORF, and SPARC

The mosaic analysis with double markers (MADM) was developed using split fluorescent proteins (split GFP and split RFP) and inter-chromosome recombination using Cre recombinase [78,79,80,81].

By crossing MADM transgenic mice with various Cre transgenic mice, single neurons were sparsely visualized using GFP, RFP, or both GFP and RFP (Figure 2). MADM enables multicolor sparse single-cell labeling (Figure 3A). The MADM method helped elucidate the genetic functions of Lis1 and Ndel1 in neuronal migration [82].

The strategy using mononucleotide repeat frameshift (MORF) is based on stochastic frameshift mutations, which are rare [83]. 

In these mice, a frameshift mutation was inserted into the N-terminus of the fluorescent reporter mNeonGreen-F, which contains a frameshift mutation that is designed to be expressed in a Cre-dependent manner. Therefore, when crossing MORF transgenic mice with various Cre transgenic mice, single neurons were sparsely visualized using mNeonGreen-F in the brain (Figure 2). 

The SPARC strategy is based on differences in recombination efficacy between the recombinase recognition sequence and its truncated mutant sequence [84]. Using the SPARC strategy, single neurons were sparsely visualized in the brains of certain transgenic flies (Figure 2).

## 10. SLENDR

The SLENDR method was developed by applying CRISPR-Cas9 technology to rodent brains [85,86,87]. The efficacy of gene editing using CRISPR-Cas9 is very low in vivo in the brain. Therefore, the insertion of small tags, such as Hemagglutinin (HA), into various endogenous genes using CRISPR-Cas9, results in the sparse expression of endogenous genes conjugated with the small tag [87,88]. The morphology of sparse single neurons was visualized using immunohistochemistry with an antibody against a small tag (Figure 2). However, it is necessary to select highly expressed endogenous proteins, such as actin or CaMKII, for the small tag’s insertion site to visualize the morphology of neurons [87,88]. Using the SLENDR method enables easy examination of synaptic loss in aged mice by visualizing postsynaptic spines [88]. Rodent brains are densely packed with many neurons; hence, performing single-cell immunohistochemistry in the brain region is challenging. The SLENDR method is unique and enables the sparse single-cell immunohistochemistry of endogenous proteins in the brain.

## 11. Functional Analysis Using Single-Cell Gene Knockout

To examine gene functions, a seminal method known as gene knockout was developed by Mario Capecchi [6]. The first-generation methodology of gene knockout was applied to the entire bodies of animals, but examining the gene functions in the local areas of neural circuits was challenging (Figure 4B). The second-generation region-specific gene knockout methods were developed using region-specific Cre transgenic mice and floxed knockin mice with two loxP sequences flanking the target genes [89,90,91,92]. These methods induced region-specific knockout of target genes and enabled the examination of the gene functions in some local regions. Furthermore, to examine the gene functions in the context of cell–cell local interactions and neural circuits, third-generation gene-knockout and single-cell gene knockout methods have been developed (Figure 4C) [11,12]. In single-cell gene knockout methods, Cre recombinase is sparsely delivered into single cells using a biolistic gene gun or transgenic virus. In one of the pioneering studies on single-cell gene knockout methods, BDNF was sparsely deleted in organotypic slice cultures of the visual cortex [12]. At the same time, BDNF was shown to have local positive functions in the synaptic formations of inhibitory neurons [12]. Furthermore, research using the single-cell gene knockout method has uncovered the local functions and developmental processes of various genes. NAPS4 has been revealed to regulate the inhibitory synaptic formation in the hippocampus [93,94]. The functional roles of NMDA receptors in the developmental processes of synapse and synaptic formation were discovered [74,95]. The subunit composition of AMPA receptors and the synaptic targeting mechanism of AMPA receptors have been revealed [96,97,98]. Some mechanisms underlying synaptic plasticity and memory have also been revealed [99,100]. The functional roles of AMPA and NMDA receptors in neuronal morphology and inhibitory synaptic formation have been reported [101]. The functional roles of LGI1 and adam22 in synaptic maturation have been previously reported [102]. Thus, biolistic gene gun, in utero electroporation, drug-activatable Cre system, photo-activatable Cre system, Supernova, BATTLE 2.0, MADM, and SLENDR are applicable for single-cell gene knockout (Figure 2).

## 12. Single-Cell Silencing and Activation

The co-expression of ion channels using a single-cell labeling vector would enable the examination of the effect of neuronal activity modulation at the single-cell level (Figure 5A,F). Neuronal activity is essential for neural development and plasticity [13], and previous studies have shown that single-cell selective neuronal suppression differs from global suppression. For instance, the focal application of an acetylcholine receptor blocker, α-bungarotoxin, induced synapse loss at the neuromuscular junction, whereas global application did not [103]. Single-cell labeling, combined with whole-cell patch-clamp recordings, revealed that activity suppression led to a reduction in synaptic inputs to hippocampal neurons [104] and dendrite arborization [105]. Thus, single-cell activity-modulating techniques are quite useful, and numerous innovative papers have been published. In an initial trial, G protein-gated inward rectifier K+ channels (GIRK) were overexpressed using the adenoviral vector, and this overexpression inhibited potential firing action [106] (Figure 5B). GIRK channels are activated by the stimulation of G protein-coupled receptors (GPCR), such as GABA_B_ and metabotropic glutamate receptors, resulting in the efflux of potassium ions and hyperpolarization of the cell membrane potential. Currently, three methods are commonly used to modulate neuronal activity, each of which differs in the duration of modulation and manner of stimulation. 

First, Kir2.1 is often used as a tool for suppressing neuronal activity (Figure 5C). as it is a strongly inwardly rectifying K^+^ ion channel [104,107]. The Kir2.1 ion channel does not require stimulation by GPCR and constantly conducts the K^+^ current at resting and modestly depolarized potential, resulting in constant suppression. In fact, viral expression of Kir2.1 suppressed neuronal activity and decreased firing, with a hyperpolarizing shift in the resting membrane potential and shunting effect [107]. In addition, Kir2.1 is substantially not toxic, whereas weakly inwardly rectifying the K^+^ channel, ROMK, induces apoptosis [108]. This lack of toxicity is probably attributed to the strong inward rectification, which reduces the efflux of K^+^ ions, K^+^ loss [108], and ATP consumption [109]. Viral vector-mediated expression of Kir2.1 affects the phenotype of neurotransmitter choice [110] and the survival of newborn neurons [111] indicating the significant role of neuronal activity in development. Conversely, neuronal activity can be increased through the expression of the bacterial Na^+^ channel, NaChBac [111] (Figure 5G). Although the I-V relationship of NaChBac is similar to that of mammalian TTX-sensitive Na^+^ channels, a key feature of a neuronal activator is its extremely long opening duration of up to several hundred milliseconds [112]. Indeed, retroviral vector-mediated expression of NaChBac increases the survival rate of newborn neurons [111]. 

Second, designer receptors exclusively activated by designer drug (DREADD)-based chemogenetics allow neuronal activity to be modulated in a drug-induced manner (Figure 5D). Since GIRK channels are stimulated by GPCRs, the overexpressed GIRK channels open with stimulation by endogenous neurotransmitters, such as GABA, making it impossible to control the timing of suppression. To rectify this, DREADD-based chemogenetics have been developed. DREADDs are genetically mutated GPCRs that do not react with the original endogenous ligands and are activated only by artificial ligands [112] (Figure 5D). Therefore, neuronal excitability can only be suppressed when a specific ligand for DREADD is administered. DREADD technology can also activate neurons through a mutated muscarinic receptor, hM3Dq, which stimulates the Gαq protein. The stimulation can inhibit outward KCNQ channel current, which is known as M current, and lead to neuronal activation [112] (Figure 5H).

Third, optogenetics, a method for modulating neuronal activity with light-responsive proteins, was reported in 2005. These are light-responsive proteins, such as channelrhodopsin and halorhodopsin, which are a cation channel and ion pump, respectively (Figure 5I,E). These proteins can depolarize or hyperpolarize a neuron through irradiation of the target neurons using light of specific wavelengths, leading to millisecond-scale modulation of neuronal excitability. Optogenetics successfully modulates neuronal activity both in vitro and in vivo [113]. Although optogenetics is a technique with high temporal and spatial specificity, it requires the implantation of laser optic fibers. 

Each of these three modulation methods has unique characteristics, which should be considered when choosing the method to use along with single-cell labeling. For example, the expression of Kir2.1 is suitable for the examination of chronic effects, DREADD for modulation on an hour-to-minute basis, and optogenetics for modulation on a second-to-millisecond basis.

## 13. Discussion

Sparse single-cell labeling methods are essential to understand the structure and function of the brain. In addition, the rodent brain is genetically programmed to develop from a fertilized egg to a complex neuronal structure encompassing the whole brain and the entire body [9]. Sparse single-cell labeling methods are helpful tools for examining the mechanisms underlying the developmental processes and programs [114,115,116]. Using single-cell labeling methods, various morphologies of presynaptic axonal terminals, dendrites, postsynaptic spines, and cell bodies can be visualized, monitored, and traced at various developmental stages. In addition, sparse single-cell labeling methods and strategies also visualize the morphologies of non-neuronal cells such as microglia, astrocytes, and oligodendrocytes [83,117].

Furthermore, applying single-cell labeling methods to global gene-knockout mice for various genes will enable the analysis of gene functions in cell morphology and synaptic formation, which are not detectable with ordinary methods. 

The use of global gene knockout methods to examine the local interactions of proteins and their functions is challenging [12]. Sparse single-cell gene knockout methods enable the dissection of the local functions of releasable proteins. As many genes have essential roles in developmental stages, global gene knockout of such genes often results in lethal phenotypes and prevents the examination of gene functions. In such cases, the single-cell gene knockout method is one of the best approaches to overcome the challenge and examine the functions of these genes. Studies using single-cell gene knockout methods must first compare gene knocked-out single neurons with control neurons. Therefore, multicolor sparse single-cell labeling methods are appropriate for these purposes (Figure 3). Biolistic gene guns, in utero electroporation, BATTLE 2.0, MADM, and SLENDR methods can achieve multicolor sparse single-cell labeling and single-cell gene knockout (Figure 2). If used in tandem with the revolutionary CRISPR/Cas9 technologies [85,86,87,88], single-cell gene knockout methods would become standard tools in various life science fields. As sparse single-cell labeling methods can potentially be applied in studies of various diseases using disease model mice, these methods will fundamentally contribute to understanding various disease mechanisms [88]. In the future, the new invention of revolutionary single-cell labeling strategies that can be applied to human postmortem brains is anticipated. This will accelerate the understanding of various neural circuits and neural cell types in the human brain and fundamentally contribute to life science, medical science, and drug discovery. Single-cell labeling enables the comparison of the phenotype and morphology of single neurons. Particularly, the neuronal activities that are genetically modulated can be compared with that of the neighboring non-modulated or differently modulated cells. As mentioned earlier, single-cell modulation is different from global modulation and is more comparable to in vivo conditions, where the neurons are competitive with each other [118]. Furthermore, BATTLE2.0 and other multicolor single-cell labeling techniques are expected to reveal the relationship between increased or decreased neural activity, neuronal phenotype, and neural connections. Additionally, the role of the pattern and timing of firing in the neuronal phenotype needs to be clarified [119], as synaptic plasticity depends on small differences in the timing of presynaptic and postsynaptic spikes [120]. Finally, single-cell labeling is a promising technique to elucidate the co-expression of different types of ion channels.

## 14. Conclusions

We have summarized the features of various sparse single-cell labeling methods. The non-transgenic staining, e.g., Golgi, HRP, and biocytin, significantly contributed to the visualization of single neuron morphologies. With the development of GFP and related fluorescent proteins, single-cell transgenic technologies have significantly advanced research in this area. Especially the BATTLE 2.0 technique enabled the multicolor and mutually exclusive single-cell labeling and transgene expressions. This technique indeed allowed the high-resolution imaging of the fine structures of both pre-and post-synapses. The expression of recombinases and gene editing enzymes can achieve single-cell gene knockout, which uncovers the functional role of the gene of interest. The co-expression of ion channels can achieve single-cell activity manipulation, of which the effect differs from the global manipulation. The contribution of this technology to neuroscience research is expected to be even more significant in the future.

## Figures and Tables

**Figure 1 biology-12-00321-f001:**
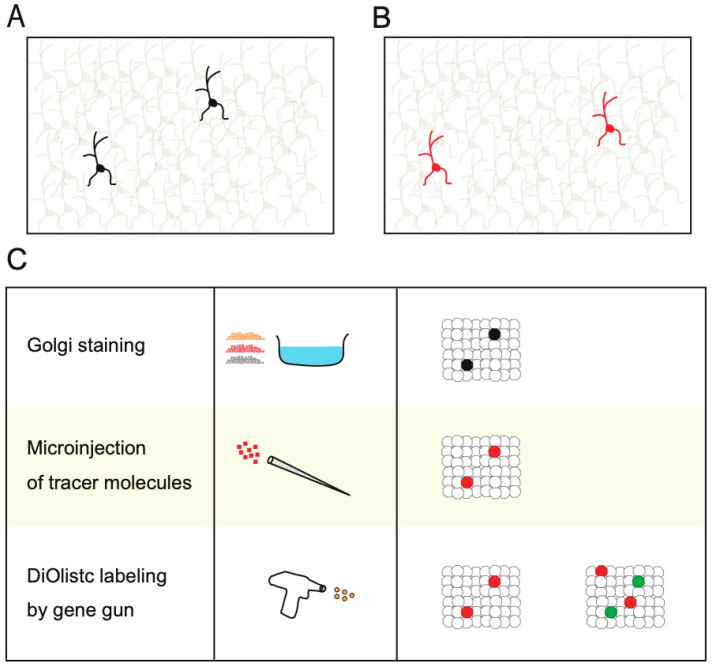
Sparse single-cell labeling using non-transgenic methods. (**A**) Illustration showing sparse single-cell labeling using the Golgi staining method. (**B**) Illustration showing the sparse single-cell labeling using microinjection of tracer molecules and DiOlistic labeling using gene gun. (**C**) Illustrations showing the characteristics of non-transgenic single-cell labeling methods. The right panel shows illustrations of mono-color and multicolor single-cell labeling.

**Figure 2 biology-12-00321-f002:**
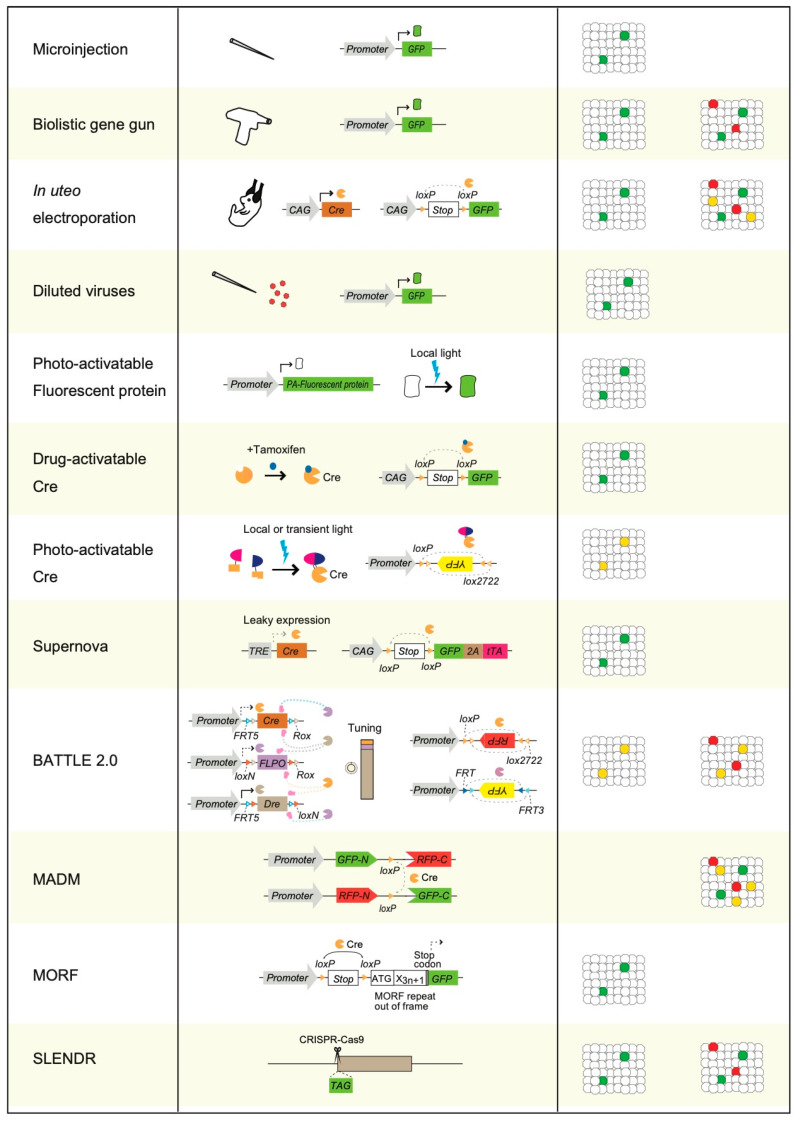
Transgenic methods and strategies for sparse single-cell labeling. Illustrations showing the characteristics, representative frameworks, and representative transgenic constructs of transgenic single-cell labeling methods and strategies. The right panel shows illustrations of mono-color and multicolor single-cell labeling.

**Figure 3 biology-12-00321-f003:**
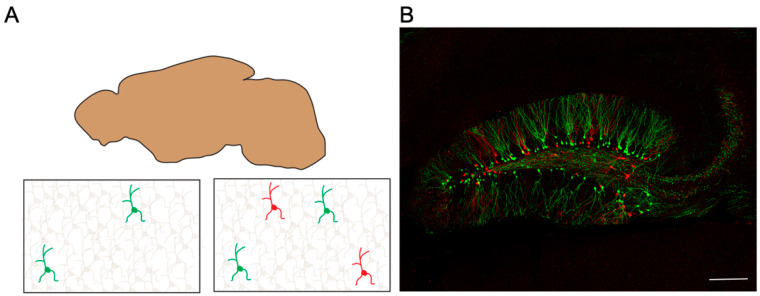
Multicolor sparse single-cell labeling using transgenic strategies. (**A**) Illustrations showing mono- and multicolor sparse single-cell labeling. (**B**) A representative image of multicolor and mutually exclusive sparse single-cell labeling using BATTLE 2.0. (Modified from Kohara K et al., 2020 [63]). The green color shows yellow fluorescent protein (YFP) fluorescence, and the red color shows mCherry’s fluorescence in the mouse hippocampus.

**Figure 4 biology-12-00321-f004:**
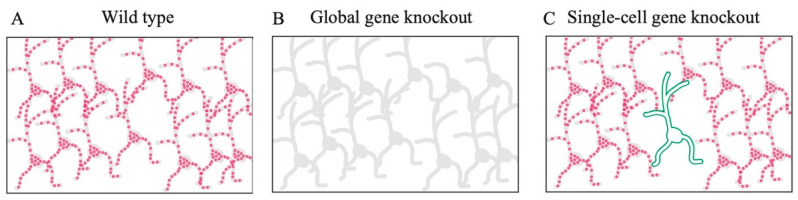
Illustrations comparing global-and single-cell gene knockout. (**A**) The illustration shows a representative brain region of wild-type mouse. Red particles represent target proteins. (**B**) The illustration shows a global gene-knockout mouse. (**C**) The illustration shows a single-cell gene knockout in a representative region. The green color indicates the neuron in which the target protein was specifically knocked out. Green fluorescent protein (GFP) visualizes the neuronal morphology of this neuron, whereas the neighboring neurons show normal expression of the target proteins. Red particles represent target proteins.

**Figure 5 biology-12-00321-f005:**
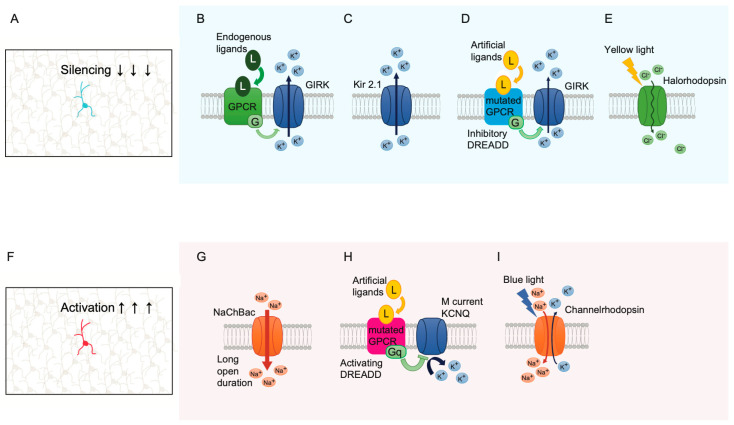
Illustrations of ion channels applied for single-cell silencing and activation. The expression of the following ion channels enables the modulation of neuronal activity at the single-cell level. (**A**) The illustration shows single-cell silencing in the brain. (**B**) The illustration of the initial silencing trial, in which GIRK channels were overexpressed. The binding of endogenous ligands to GPCR activates the GIRK, resulting in neuronal silencing. (**C**) Kir2.1 channels always conduct outward current except when strongly depolarized, leading to silencing. (**D**) DREADD-based silencing: the mutated GPCRs are activated only by the administration of artificial ligands, not by endogenous ligands, leading to time-controllable silencing. (**E**) Halorhodopsin is activated with the yellow light and conduct of inward currents of Cl ion, resulting in precise temporal control of neural silencing. (**F**) The illustration shows single-cell activation in the brain. (**G**) NaChBac channels, which open voltage-dependently, conduct extremely long-lasting inward currents. (**H**) The ligand binding to the hM3Dq receptor stimulates Gαq protein that inhibits the KCNQ channel, leading to neuronal activation. (**I**) Channelrhodopsin is activated with the blue light and conducts inward and outward currents, resulting in precise temporal control of neural activity.

## Data Availability

Data sharing not applicable.

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
