# Peer review of "Single-Cell Labeling Strategies to Dissect Neuronal Structures and Local Functions"

_biology, 2023, doi:10.3390/biology12020321_

Round 1

Reviewer 1 Report

This is a useful review on current strategies on applications for single-cell labeling in the nervous system. 

Comments:

Keywords.  methods; strategies are not useful as keywords. Perhaps use single-cell labeling methods since strategies is used in title.  Also, "activating" should be "activation".

Keywords.  methods; strategies are not useful as keywords. Perhaps pick use single-cell labeling methods since strategies is used in title.  Also, "activating" should be "activation".

Section 2. 

para 1.  Camillo Golgi spelling.

Suggest not duplicating text used in Introduction.

In Figure 2 and throughout text.  Change in uteo to in utero.

Section 4. para 1, last sentence. Remove “in some cases”.

Section 5. last sentence.  Statement about photoactivation and photoconversion use in small cells and not large neurons should be clarified.  The methods are commonly used to small and large neurons. The authors might want to emphasize the preference for local photoactivation (dendritic arbors) or differences between cell culture and tissue use.

Section 6, para. 1. Explain what is meant by light-sensitive proteins. What kind of light-sensitive dimerization system?

Section 10. Line 234.  Explain what is meant by “…need to be selected for the insertion site…”

Capital S in section title Single cell silencing….

Section 12. line 362-363. Explain who the strategies would be applied to post-mortem human tissue.

Author Response

Response to reviewer 1  

This is a useful review on current strategies on applications for single-cell labeling in the nervous system. 

Comments:

Keywords.  methods; strategies are not useful as keywords. Perhaps use single-cell labeling methods since strategies is used in title.  Also, "activating" should be "activation".

Keywords.  methods; strategies are not useful as keywords. Perhaps pick use single-cell labeling methods since strategies is used in title.  Also, "activating" should be "activation".

According to the editor’s request, we added simple summary and conclusion.

We agree your suggestions. We modified those sentences.

Section 2.  

para 1.  Camillo Golgi spelling. 

We corrected Camillo Golgi spelling.

Suggest not duplicating text used in Introduction. 

We deleted duplicated sentences.

In Figure 2 and throughout text.  Change in uteo to in utero.

We modified those words.

Section 4. para 1, last sentence. Remove “in some cases”.

We modified the sentence.

Section 5. last sentence.  Statement about photoactivation and photoconversion use in small cells and not large neurons should be clarified.  The methods are commonly used to small and large neurons. The authors might want to emphasize the preference for local photoactivation (dendritic arbors) or differences between cell culture and tissue use. 

We are sorry for those confusing sentences, and we deleted those sentences.

Section 6, para. 1. Explain what is meant by light-sensitive proteins. What kind of light-sensitive dimerization system? 

We added representative light-sensitive dimerization systems. In addition, we explained the other single-chain photoactivatable Cre system using LOV domain of protein VVD.

Section 10. Line 234.  Explain what is meant by “…need to be selected for the insertion site…”

We modified the sentence.

Capital S in section title Single cell silencing….

We modified the sentence.

Section 12. line 362-363. Explain who the strategies would be applied to post-mortem human tissue. 

 We modified the sentence.

Reviewer 2 Report

The manuscript titled "Single-cell labeling strategies to dissect neuronal structures and local functions" is detailed and well-illustrated review. This manuscript provides a comprehensive review of commonly used single-cell labeling strategies for the study of neural structure and local function, and provides an outlook on the future applications of single-cell labeling techniques. I only have minor comments below:

In lines 34 and 60-61, the authors state that Golgi stain is incompatibility for combinational use with immunohistochemistry. However, several publications have used Golgi-Cox-stained tissue for immunostaining with impressive results. They may need to explicitly state in which condition Golgi stain is incompatible or compatible with immunohistochemistry. In addition, the "single-cell silencing and activation" section in lines 276-332 provides a review of the various modes of neuronal manipulation, and it would be better to add a corresponding pattern diagram in this section to show it more clearly.

Author Response

Response to reviewer 2

The manuscript titled "Single-cell labeling strategies to dissect neuronal structures and local functions" is detailed and well-illustrated review. This manuscript provides a comprehensive review of commonly used single-cell labeling strategies for the study of neural structure and local function, and provides an outlook on the future applications of single-cell labeling techniques. I only have minor comments below:

In lines 34 and 60-61, the authors state that Golgi stain is incompatibility for combinational use with immunohistochemistry. However, several publications have used Golgi-Cox-stained tissue for immunostaining with impressive results. They may need to explicitly state in which condition Golgi stain is incompatible or compatible with immunohistochemistry. In addition, the "single-cell silencing and activation" section in lines 276-332 provides a review of the various modes of neuronal manipulation, and it would be better to add a corresponding pattern diagram in this section to show it more clearly.

Thank you for your suggestions. We deleted the part “ Golgi staining is incompatibility for combinational use with immunohistochemistry.”

We added new figure 5 and the sentence which explains activating the DREADD system in the single-cell silencing and activation”.